# TongGuanWan Alleviates Doxorubicin- and Isoproterenol-Induced Cardiac Hypertrophy and Fibrosis by Modulating Apoptotic and Fibrotic Pathways

**DOI:** 10.3390/ijms251910573

**Published:** 2024-09-30

**Authors:** Jung-Joo Yoon, Ai-Lin Tai, Hye-Yoom Kim, Byung-Hyuk Han, Sarah Shin, Ho-Sub Lee, Dae-Gill Kang

**Affiliations:** 1Hanbang Cardio-Renal Syndrome Research Center, Wonkwang University, 460, Iksan-daero, Iksan 54538, Republic of Korea; mora16@naver.com (J.-J.Y.); 775452499@gg.com (A.-L.T.); hyeyoomc@naver.com (H.-Y.K.); arum0924@naver.com (B.-H.H.); 2College of Oriental Medicine and Professional Graduate School of Oriental Medicine, Wonkwang University, 460, Iksan-daero, Iksan 54538, Republic of Korea; 3KM Science Research Division, Korea Institute of Oriental Medicine, 1672, Yuseong-daero, Yuseong-gu, Daejeon 34054, Republic of Korea; s.sarah@kiom.re.kr

**Keywords:** TongGuanWan, doxorubicin, cardiac hypertrophy, fibrosis, apoptosis, heart failure

## Abstract

Heart failure, a major public health issue, often stems from prolonged stress or damage to the heart muscle, leading to cardiac hypertrophy. This can progress to heart failure and other cardiovascular problems. Doxorubicin (DOX), a common chemotherapy drug, and isoproterenol (ISO), a β-adrenergic agonist, both induce cardiac hypertrophy through different mechanisms. This study investigates TongGuanWan (TGW,), a traditional herbal remedy, for its effects on cardiac hypertrophy and fibrosis in DOX-induced H9c2 cells and ISO-induced mouse models. TGW was found to counteract DOX-induced increases in H9c2 cell surface area (*n* = 8, *p *< 0.01) and improve biomarkers like ANP (*n* = 3, *p *< 0.01)) and BNP (*n* = 3, *p *< 0.01). It inhibited the MAPK pathway (*n* = 4, *p *< 0.01) and GATA-4/calcineurin/NFAT-3 signaling, reduced inflammation by decreasing NF-κB p65 translocation, and enhanced apoptosis-related factors such as caspase-3 (*n* = 3, *p *< 0.01), caspase-9 (*n* = 3, *p *< 0.01), Bax (*n* = 3, *p *< 0.01), and Bcl-2 (*n* = 3, *p *< 0.01). Flow cytometry showed TGW reduced apoptotic cell populations. In vivo, TGW reduced heart (*n* = 8~10, *p *< 0.01), and left ventricle weights (*n* = 6~7), cardiac hypertrophy markers (*n* = 3, *p *< 0.01), and perivascular fibrosis in ISO-induced mice, with Western blot analysis confirming decreased levels of fibrosis-related factors like fibronectin, α-SMA (*n* = 3, *p *< 0.05), and collagen type I (*n* = 3, *p *< 0.05). These findings suggest TGW has potential as a therapeutic option for cardiac hypertrophy and fibrosis.

## 1. Introduction

Cardiovascular disease remains a leading cause of mortality among adults and the elderly, encompassing a range of conditions such as atherosclerosis, hypertension, cardiomyopathy, congestive heart failure, myocardial infarction, and diabetes mellitus [1,2]. One early and significant manifestation of cardiovascular disease is cardiac hypertrophy, which represents an adaptive response of cardiac muscle cells to various pathological stimuli, including exogenous β-adrenergic receptor agonists like isoproterenol (ISO) [3]. This condition is characterized by an increase in cell size, protein synthesis, and the expression of fetal genes such as atrial natriuretic peptide (ANP), brain natriuretic peptide (BNP), β-myosin heavy chain (β-MHC), and skeletal α-actin [4,5]. Specifically, ANP is predominantly secreted by atrial myocytes, while BNP is mainly produced by ventricular myocytes, both of which are released in elevated quantities during cardiac hypertrophy [6].

The intracellular mitogen-activated protein kinase (MAPK) signaling pathway is crucial in the development of cardiac hypertrophy in response to various stimuli [7]. MAPKs and nuclear factor kappa-light-chain-enhancer of activated B (NF-κB) signaling pathways are important for cardiac fibrosis progression [8,9]. Activation of the MAPK signaling pathway can further activate NF-κB, leading to nuclear translocation of NF-κB for target gene transcription activation. Previous studies have highlighted NF-κB’s role in fibroblast proliferation and collagen expression, particularly through pathways involving angiotensin II and tumor necrosis factor-alpha (TNF-α) [10]. Moreover, calcineurin–nuclear factor of activated T cells (NFAT) signaling may integrate with another pathway, such as protein kinase C or MAPKs, to coordinate the hypertrophic response [11].

Calcineurin is a serine/threonine-specific phosphatase that is uniquely activated by sustained elevations of intracellular calcium and is associated with hypertrophic signaling factors [12,13,14]. Calcineurin and NFAT are known for activation of the fetal gene program in response to hypertrophic stimuli, acting as essential effectors during the formation of cardiac hypertrophy [15]. In cardiac hypertrophy, calcium-dependent phosphatase calcineurin dephosphorylates the nuclear factor of NFAT-3 transcription factor, allowing it to translocate to the nucleus, leading to overexpression of atrial natriuretic (ANP) and b-type natriuretic peptide (BNP) [14]. Activated calcineurin binds directly to the NFAT-3 transcription factor in the cytoplasm and dephosphorylates it. Dephosphorylated NFAT-3 further interacts with the GATA binding protein 4 (GATA-4) transcription factor, allowing translocation to the nucleus to form a complex that participates in the development of myocardial hypertrophy. In addition to NFAT-3, GATA-4 is a transcription factor that plays an important role in the downstream part of the activated calcineurin signaling pathway, mediating the cardiac hypertrophy response [16,17]. GATA-4 is a cardiac-restricted zinc finger protein that participates in cardiac development and related gene expression. Overexpression of GATA-4 leads to myocardial hypertrophy in vivo by regulating the expression of various genes associated with cardiac hypertrophy and heart failure, including ANP, BNP, α-myosin heavy chain (α-MHC), and β-MHC [18]. Therefore, the calcineurin/NFAT/GATA4 pathway functions as an essential effector during cardiac hypertrophy formation.

Cardiac hypertrophy results in various types of cardiomyocyte cell death (e.g., necrosis, apoptosis, and autophagy). Pathological cardiac hypertrophy and myocardial dysfunction are strongly linked to cardiac myocyte apoptosis, which plays an important role in cardiac remodeling resulting in hypertrophy leading to sudden cardiac death [19,20]. Given that apoptosis is a critical factor in the pathogenesis of many heart diseases, targeting or preventing cardiac apoptosis represents a potential therapeutic strategy for heart failure [21,22].

Cardiac fibrosis is one of the major pathological changes in the heart caused by the accumulation of extracellular matrix components such as collagen and excessive proliferation of cardiac fibroblasts. It is a major cause of cardiac stiffness, arrhythmias, and heart failure [23,24]. Although the exact mechanisms regulating the process of cardiac fibrosis are incompletely understood, the transforming growth factor-β (TGF-β)/Smads signaling pathway is known to play an important role in cardiac fibrosis. TGF-β1 is a representative fibrogenic factor and plays a role in various diseases such as cardiac hypertrophy as well as cardiac abnormalities, cardiac fibrosis, cardiac dysfunction, and chamber remodeling. TGF-β can induce Smad2/Smad3 (downstream mediator) phosphorylation, which in turn can stimulate cardiac fibrosis development. Smad2/3 activation affects the expression of various profibrotic genes in relation to collagens, various proteoglycans, integrins, connective tissue growth factor (CTGF), and matrix metalloproteinases (MMPs) [25]. Therefore, suppression of the TGF-β/Smad signaling pathway is a promising therapy for cardiac fibrosis [26].

TongGuanWan (TGW) is a well-known traditional Chinese herbal medicine consisting of *Anemarrhenae* Rhizoma (*Anemarrhena asphodeloides* Bunge., Zhimu), *Phellodendri chinensis* Cortex (*Phellodendron chinense* Schneid., Huangbai) and *Cinnamomum cassia* (*Cinnamomum cassia* Presl, Rougui). TGW was first documented in ‘LanShiMiCang’, a famous ancient medical treatise written by Li Gao during the Yuan Dynasty, where it was described as a prescription for treating diseases with ‘yin and fever’ syndrome [27,28]. In China and other countries, the three main components of TGW are frequently used to treat diabetes or its complications. In a previous study, TGW lowered blood glucose and improved glucose tolerance in db/db mice and exhibited anti-inflammatory activity by inhibiting interleukin (IL) and regulating the NF-κB pathway [27]. However, the effect of TGW on heart dysfunction via cardiac hypertrophy has not been demonstrated yet. This study investigated the protective effect of TGW on cardiac hypertrophy through the regulation of apoptosis and fibrosis.

## 2. Results

### 2.1. Chemical Characterization of TongGuanWan

In total, 23 compounds were identified by ultra-performance liquid chromatography employing quadrupole exactive Orbitrap mass spectrometry (UPLC/QE Orbitrap MS) analysis. Of these, 9 compounds—4-hydroxy cinnamic acid, catapol, coumarin, ferulic acid, mangiferin, morroniside, neomangiferin, palmatine, and protocatechuic acid—were verified based on the authentic compounds, and 14 compounds were tentatively identified according to the retention time, exact MS, and MS/MS fragments. In particular, chemical markers of each herb, including ferulic acid, 4-hydroxy cinnamic acid, coumarin, mangiferin, neomangiferin, berberine, and palmatine, were identified in TGW (Figure 1). Details about the identified compounds are listed in the Appendix A.

### 2.2. Effect of TongGuanWan on Cardiomyocyte Hypertrophy in Doxorubicin (DOX)-Treated H9c2 Cells

For the evaluation of the effect of TGW pretreatment on DOX-induced cytotoxicity, H9c2 cells were pretreated for 60 min with TGW (1–10 μg/mL). Then, the medium was changed and cells were treated with DOX (1 μM) for 24 h. TGW amplified the reduction in cell viability induced by DOX in a manner that depended on the concentration of the TGW pretreatment (Figure 2A). Additionally, DOX was found to increase the surface area of the H9c2 cells by approximately 2.64-fold compared with the control group (*p *< 0.01). However, pretreatment with TGW 10 μg/mL blocked the DOX-induced increase in cell size (3.03 ± 0.17 versus 1.83 ± 0.12 (*p *< 0.01), Figure 2B). ANP, BNP, β-MHC, and MLC-2v are known as representative cardiac hypertrophy markers. To determine whether TGW suppressed these cardiac hypertrophy markers, we performed Western blotting and real-time reverse transcription–polymerase chain reaction (RT-PCR). As shown in Figure 3, protein expression of ANP, BNP, β-MHC, and myosin light chain-2v (MLC-2v) was significantly increased by DOX. However, treatment with TGW significantly decreased the expression of hypertrophy biomarker proteins (Figure 3A,B). Similarly, TGW significantly inhibited DOX-induced ANP, BNP, β-MHC, and MLC-2v mRNA levels in H9c2 cells (Figure 3C). These results suggest that TGW ameliorated DOX-induced cardiomyocyte size reduction and cardiomyocyte hypertrophy.

### 2.3. Effect of TongGuanWan on the Calcineurin/NFAT and GATA4 Pathway 

Calcineurin is a key signaling mediator implicated in the modulation of hypertrophic signaling pathways. Pathological cardiac hypertrophy and heart failure are characterized by calcineurin overexpression in the heart and activation of the calcineurin/NFAT and GATA4 pathway. In this study, TGW treatment was found to diminish the DOX-induced increase in calcineurin protein levels in H9c2 cells (Figure 4A). Additionally, we analyzed the levels of NFAT-3 in both the cytoplasm and nucleus following DOX exposure. As illustrated in Figure 4B, TGW pretreatment led to a dose-dependent reduction in the nuclear NFAT-3 levels, which was associated with a concurrent increase in cytosolic levels. 

We examined activation of transcription factor GATA-4, which is implicated in the development of cardiac hypertrophy, following the administration of DOX. The results showed that the level of phosphorylated GATA-4 was highly influenced by DOX. However, TGW effectively prevented DOX-induced p-GATA-4 protein expression (Figure 4C). In addition, an immunofluorescence assay was performed to determine the effect of TGW on DOX-induced nuclear translocation of GATA-4 phosphorylation. As shown in Figure 4D, pretreatment with TGW inhibited the nuclear localization activation of GATA-4 compared with DOX-only treatment. These results suggest that TGW improves cardiac hypertrophy by regulating the calcineurin/NFAT and GATA-4 pathway, which is a transcriptional regulator for the generation of cardiac hypertrophy.

### 2.4. Effect of TongGuanWan on the Expression of MAPK Signaling Pathways and NF-kB Expression

In our investigation of the effects of TGW on MAPK signaling pathways involved in cardiac hypertrophy and heart failure, we observed the following. Figure 5 illustrates that DOX treatment increased the phosphorylation levels of JNK, ERK, and p38 MAPK. However, these phosphorylated levels were significantly reduced with high-dose TGW treatment. Figure 5B demonstrates that DOX treatment led to a marked increase in NF-κB expression, which was subsequently decreased by treatment with TGW at a concentration of 10 μg/mL. Additionally, our study included an immunofluorescence assay to further verify the effect of TGW on NF-κB nuclear translocation in H9c2 cells exposed to DOX. Figure 5C shows that TGW treatment notably enhanced the nuclear localization of NF-κB. These results suggest that TGW inhibits cardiac hypertrophy by blocking MAPK signaling pathways and NF-κB expression.

### 2.5. The Effect of TongGuanWan on Cell Apoptosis Marker

To clarify the effects of TGE on DOX-induced cardiac apoptosis, we performed Western blotting and RT-PCR. As indicated in Figure 6A, DOX treatment resulted in an upregulation of pro-apoptotic markers, including Bcl-2-associated X protein (Bax) and the cleaved forms of caspases-3, -8, and -9. Conversely, there was a noticeable reduction in the levels of Bcl-2, a key anti-apoptotic protein. Treatment with TGW prior to DOX exposure led to notable decreases in the levels of Bax and cleaved caspases-3, -8, and -9, and a corresponding increase in Bcl-2 expression in H9c2 cells. Additionally, while DOX treatment caused an upregulation of Bax and caspase-3 mRNA, TGW effectively counteracted this effect. High-dose TGW treatment further resulted in a significant increase in Bcl-2 mRNA levels (Figure 6B In this study, we further assessed the impact of TGW on DOX-induced cardiac apoptosis using annexin V-fluorescein isothiocyanate (FITC) and propidium iodide (PI) dual staining in conjunction with flow cytometry. Figure 6C demonstrates that TGW treatment significantly reduced the apoptosis rate compared with treatment with DOX alone. These observations suggest that TGW effectively mitigates DOX-induced cardiac apoptosis by modulating apoptosis-related factors, thereby contributing to the improvement of heart failure associated with cardiac hypertrophy.

### 2.6. Effect of TongGuanWan on Cell Fibrosis Marker

To evaluate the potential therapeutic effect of TGW on DOX-induced fibrosis in H9c2 cells, we conducted a comprehensive analysis of fibrosis-related protein expression levels, including fibronectin, collagen I, α-smooth muscle actin (α-SMA), TGF-β1, and p-Smad3. As shown in Figure 7, treatment with DOX significantly upregulated the expression of these fibrosis-related proteins, indicating a robust induction of fibrotic processes within the cells. In contrast, treatment with TGW at concentrations of 5 µg/mL or higher resulted in marked reductions in the expression of collagen I, α-SMA, TGF-β1, and p-Smad3, all of which are key markers involved in fibrosis progression. Interestingly, fibronectin expression did not exhibit the same level of reduction at lower concentrations of TGW. However, a noticeable decrease in fibronectin levels was observed when the concentration of TGW reached 10 µg/mL. These findings suggest a dose-dependent antifibrotic effect of TGW, where higher concentrations are required to modulate fibronectin expression. Collectively, these data highlight the capacity of TGW to effectively attenuate DOX-induced cardiac fibrosis by modulating the expression of key fibrosis-related genes and proteins.

### 2.7. The Effect of TongGuanWan on Cardiac Hypertrophy in ISO-Treated Mice

To identify whether TGW had a preventive effect on the development of cardiac hypertrophy, we administered TGW to mice for 1 week prior to ISO injection, followed by co-administration of TGW and ISO for 1 week. As shown in Figure 8A,B, TGW administration significantly reduced the ratio of heart weight to body weight (*p *< 0.01, Figure 8A) and the ratio of left ventricular weight to body weight compared with the ISO-only group (not significant, Figure 8B). This is also evident in Figure 8C. ISO injection significantly increased heart size compared with the control group, while the increase in heart size was attenuated in the group that received TGW (200 mg/kg/day). The dose of TGW used in the animal experiments was determined based on the results of several studies and was found to be well tolerated by mice [29,30,31,32]. We next observed that TGW administration attenuated ISO-induced cardiac hypertrophy, as evidenced by wheat germ agglutinin (WGA) staining (Figure 8D). We further examined the effects of TGW on expression of ANP, BNP, β-MHC, and MLC-2v, which are markers of cardiac hypertrophy. As shown in Figure 8E, in the ISO group, there was significantly increased protein expression of cardiac hypertrophic marker genes in the left ventricle tissue. The administration of TGW significantly reduced the ISO-induced expression of ANP, BNP, β-MHC, and MLC-2v. These data suggest that TGW inhibits left ventricular hypertrophy. 

### 2.8. The Effect of TongGuanWan on Iso-Induced Cardiac Fibrosis

Fibrosis, which is characterized by accumulation of extracellular matrix proteins, usually occurs in cardiac hypertrophy. To determine whether ISO induced cardiac fibrosis, we examined staining with picro-sirius red. As shown in Figure 9A, fibrosis was observed in the ISO group. TGW pretreatment suppressed cardiac fibrosis in left ventricle sections from ISO-induced mice. We next examined the effects of TGW on expression of fibronectin, collagen I, α-SMA, and TGF-β1, which are fibrosis-related genes. As shown in Figure 9B, ISO-induced left ventricle tissue showed increases in fibronectin, collagen I, and α-SMA protein levels, which were reduced by TGW administration. ISO increased TGF-β1 and Smad3 protein expression. This increase was considerably decreased by the administration of TGW at a dose of 200 mg/kg/day. Furthermore, TGW reduced fibronectin, collagen I, and α-SMA mRNA expression in ISO-induced mice (Figure 9C). These data indicate that TGW inhibits left ventricle hypertrophy and cardiac fibrosis by directly regulating fibrosis-related gene expression.

## 3. Discussion

Heart failure is often attributed to cardiomyocyte dysfunction, as it impairs the heart’s ability to pump blood efficiently, leading to insufficient oxygen and nutrient supply to tissues [33]. Persistent cardiac hypertrophy induced by pathological stimuli leads to the development of ventricular dilatation, systolic dysfunction, and heart failure [34]. Therefore, research aimed at understanding the molecular mechanisms underlying cardiac hypertrophy is crucial for identifying therapeutic targets to prevent or treat heart failure.

TGW is a well-known traditional herbal medicine consisting of three herbal medicines: *Anemarrhenae* Rhizoma, *Phellodendri chinensis* Cortex, and *Cinnamomum cassia* [27,28]. TGW is often used to treat diabetes or its complications. However, while some of the main active ingredients of TGW have been shown to have hypoglycemic and renal protective effects, their effects on improving heart damage have not been investigated. To the best of our knowledge, the present study is the first study to examine the cardioprotective effect of TGW through improvement of cardiac hypertrophy and fibrosis.

First, the present study shows that TGW prevented cardiotoxicity, cardiac hypertrophy and apoptosis in DOX-induced H9c2 cells. Cardiotoxicity of anthracyclines, such as DOX, epirubicin, and daunomycin, is known to be a major problem in clinical practice. However, although these drugs are widely used, they are potent chemotherapy agents with serious cardiotoxicity effects. Therefore, the therapeutic use of DOX is limited due to cumulative dose-dependent cardiotoxicity in patients with cancer, including fatal cardiomyopathy. Various compounds have been used to protect the heart from DOX-induced cardiotoxicity, but there are limitations to the long-term and short-term preventive treatment of DOX-induced cardiotoxicity [35,36]. In the present study, we demonstrated that TGW significantly ameliorated DOX-induced toxicity in H9c2 cells. 

Recent studies have confirmed that DOX induces hypertrophy in H9c2 cells, which is consistent with our findings [37,38,39]. In several pathological events, including myocardial infarction and hypertension, cardiomyocytes undergo hypertrophic responses, characterized by expansion of cell size, increased protein synthesis, and stimulation of ANP and BNP [40]. Our findings confirmed that DOX induced a hypertrophic response, which correlated with increases in the myocardial hypertrophy biomarkers ANP and BNP. Conversely, TGW significantly reduced protein and mRNA levels of myocardial hypertrophy biomarkers in H9c2 cells, indicating an efficient protective effect against DOX-induced hypertrophy. In the heart, various hypertrophic stimuli are known to activate receptors or channels and trigger multiple signaling cascades, of which the best known is the intracellular MAPK pathway. Activation of the MAPK signaling pathway further activates NF-κB, leading to nuclei of NF-κB for target gene transcriptional activation. It can induce an electric potential. These MAPK and NF-κB signaling pathways play an important role in cardiac fibrosis progression [8,9]. Our results show that TGW pretreatment down-regulated the expression of MAPK and NF-κB increased by DOX. In this study, we clearly showed that TGW pretreatment prevented DOX-induced cardiac dysfunction by blocking of the MAPK and NF-kB signaling pathways.

Calcineurin is a serine/threonine-specific phosphatase that is activated by sustained increases in calcium, which directly dephosphorylates members of the NFAT transcription factor family of activated T cells in the cytoplasm, promoting translocation to the nucleus. Nuclear-activated NFAT-3 interacts with the GATA-4 transcription factor to induce myocardial hypertrophy and the expression of hypertrophic response genes such as ANP and BNP [14,41]. GATA-4 is a cardiac-restricted zinc finger protein that participates in cardiac development and related gene expression, and overexpression of GATA-4 is known to result in myocardial hypertrophy in vivo [17,42]. Our results show that TGW reduced DOX-induced calcineurin protein expression in H9c2 cells. TGW pretreatment resulted in a dose-dependent reduction in the nuclear levels of NFAT-3, accompanied by a corresponding increase in its cytosolic levels. In addition, TGW treatment reduced p-GATA-4 protein expression and inhibited the nuclear localization of GATA-4 compared with DOX-only treatment. In previous studies, calcineurin enzyme activity and protein levels were significantly upregulated in the hearts of young tropomodulin transgenic mice, a model of diastolic heart failure, and calcineurin-NFAT signaling was activated in pathological cardiac hypertrophy and heart failure [43]. Numerous other studies have demonstrated that the calcineurin/NFAT signal and GATA-4 signaling pathway plays a pivotal role in the progression of cardiac hypertrophy [44]. Accordingly, our results suggest that TGW improves cardiac hypertrophy by regulating the calcineurin/NFAT and GATA4 pathway, which is a transcriptional regulator for the generation of cardiac hypertrophy. 

Cardiac dysfunction due to acute lesions (myocardial infarction and ischemia/reperfusion) or chronic diseases of the heart (ischemia and dilated cardiomyopathy) is associated with myocardial cell death [19,20]. Cardiac apoptosis involves morphological features that include cell contraction and cytoplasmic and chromatin condensation; chronic, low-level cardiac myocyte apoptosis has been suggested as a causal factor in the pathogenesis of heart failure. In particular, increased apoptosis can be detrimental in hypertrophic cardiomyopathy, leading to dilatation and ultimately, systolic failure [44]. Expression of factors involved in apoptosis, such as Bcl-2 and caspase-3, has been suggested as an important indicator of cardiomyocyte survival probability [45]. The current results indicate that DOX treatment led to an increase in pro-apoptotic factors, including Bax and cleaved forms of caspases-3, -8, and -9, while decreasing Bcl-2 expression, an anti-apoptotic protein. In contrast, pre-treatment with TGW reduced the levels of Bax and cleaved caspases-3, -8, and -9 and elevated Bcl-2 expression in H9c2 cells. Furthermore, the effect of TGW on DOX-induced cardiac apoptosis was confirmed using annexin V-FITC/PI dual staining and flow cytometry. These findings suggest that TGW treatment mitigates DOX-induced cardiac apoptosis through the modulation of apoptosis-related factors and consequently, cardiac hypertrophy. Toxin-induced cardiomyopathy occurs through multiple mechanisms including increased oxidative stress, lipid peroxidation, DNA damage, apoptosis, and autophagy [46]. BAX activation has recently been shown to be a rate-limiting step in doxorubicin-induced cardiomyopathy [47].

Second, the present study showed that TGW prevented cardiac hypertrophy and fibrosis in the ISO-induced mouse model. ISO is a β-adrenergic agonist, one of the biologically active neurohormonal molecules. ISO-induction models similar disease progression and is associated with low animal mortality, and it has been widely used to induce cardiac hypertrophy, fibroblast activation, and cardiac fibrosis. Pathological cardiac hypertrophy, a major and independent risk factor for heart failure, is characterized by enlarged myocardial cell surface area, interstitial fibrosis, apoptosis, and release of ANP and BNP [48,49]. Previous studies have shown that ISO activates ANP and BNP, two major biomarkers of myocardial hypertrophy [50,51,52]. Our results confirmed that ISO treatment induced cardiac hypertrophy by increasing the ratio of cardiac weight to body weight, LV/BW, and myocardial cell surface area. In addition, at the protein level, it was shown to increase expression of the hypertrophic genes ANP, BNP, β-MHC, and MLC-2v. These results are consistent with the results of previous studies [53]. However, TGW inhibits cardiac hypertrophy, as observed via the decrease in protein expression of hypertrophy biomarkers such as ratio of cardiac weight to body weight, ANP, and BNP.

In cardiac hypertrophy, various genes involved in cardiac remodeling, such as those relating to α-skeletal actin, MMP, and collagen levels, are overexpressed [54]. Increased myocardial fibrosis and collagen deposition are representative features of cardiac hypertrophy [55]. TGF-β is a pleiotropic cytokine that plays a pivotal role in cardiac remodeling and fibrosis in hypertrophic cardiomyopathy [56]. As a result of picrosirius staining, the left ventricle of the control group showed a normal structure with no fibrosis, but fibrosis was observed in the ISO group. TGW pretreatment suppressed cardiac fibrosis in the left ventricular sections of ISO-induced mice. In addition, TGW down-regulated protein and mRNA expression of factors involved in fibrosis in ISO-induced mice. These in vivo results were similar in vitro. It was shown that TGW significantly suppressed DOX-induced fibrosis-related gene protein expression in H9c2 cells. Therefore, these results suggest that TGW suppresses cardiac fibrosis by directly regulating fibrosis-related gene expression. 

In conclusion, this study provides compelling evidence that TGW effectively prevented cardiac hypertrophy and fibrosis both in DOX-induced H9c2 cell models and ISO-induced mouse models of cardiac hypertrophy. This study demonstrates that TGW mitigates cardiac hypertrophy by influencing key hypertrophy markers and regulating critical signaling pathways, including calcineurin/NFAT and GATA-4. Additionally, TGW has protective effects against cardiac fibrosis through the inhibition of TGF-β/Smads signaling pathway activity. These results indicate that TGW not only reduces the structural and molecular hallmarks of cardiac hypertrophy but also prevents the progression of fibrosis, a common and detrimental feature of chronic cardiac diseases. These findings highlight the potential of TGW as a new therapeutic agent for addressing heart disease characterized by cardiac hypertrophy and fibrosis. This study shows the ability of TGW to modulate multiple pathways involved in cardiac damage, suggesting its potential for developing new strategies to manage heart-related conditions. Future research should focus on further elucidating the molecular mechanisms through which TGW exerts these effects and exploring its potential clinical applications. 

## 4. Methods

### 4.1. Chemicals

Doxorubicin (DOX, D1515) and isoprenaline (ISO, 5 gm isoprenaline hydrochloride) were purchased from Sigma-Aldrich (St. Louis, MO, USA). Penicillin–streptomycin *F*-actin, Alexa Four 488 phalloidin, and 4′,6-diamidino-2-phenylindole, dihydrochloride (DAPI) were sourced from Invitrogen, Thermo Fisher Scientific, Inc. (Waltham, MA, USA). We purchased anti-p38, anti-ERK, anti-JNK, anti-p-ERK1/2, anti-p-p38, anti-NF-κB p65, anti-Bcl-2, anti-β-actin, anti-β-MHC, anti-MLC-2v, anti-p-GATA-4, anti-GATA-4, anti-calcineurin, and anti-lamin B from Santa Cruz Biotechnology (1:1000 dilutions, Santa Cruz, CA, USA). We also procured anti-NFAT-c3, anti-caspase-3, anti-caspase-8, anti-caspase-9, anti-Bax, and anti-α-tubulin from Cell Signaling Technology (1:1000 dilutions, Danvers, MA, USA). Anti-ANP and anti-BNP were purchased from Abcam (1:1000 dilutions, Branford, CT, USA).

### 4.2. TongGuanWan Preparation

All plant materials for TGW were purchased from the Herbal Medicine Cooperative Association (Iksan, Republic of Korea). Based on their morphological characteristics, assurance of quality control for all the materials was validated according to the Korean Herbal Pharmacopoeia and botanical names were checked based on the Plant List (www.theplantlist.org), accessed on 30 September 2024. TGW was prepared as an extract from a mixture of three dried herbs including *Anemarrhenae* Rhizoma (*Anemarrhena asphodeloides* Bunge., Zhimu)*, Phellodendri chinensis* Cortex (*Phellodendron chinense* Schneid., Huangbai) and *Cinnamomum cassia* (*Cinnamomum cassia* Presl, Rougui) (total weight = 273.75 g). The herbal TGW mixture was boiled with 1500 mL of distilled water (DW) for 2 h using a rotary evaporator and to obtain the dried extract. A total of 27.1 g of dried extract was yielded (extract yield: 79.1 g). The dried extract was stored at −70 °C and dissolved in PBS before use. The chemical compounds in TGW were analyzed using UPLC/QE Orbitrap MS. Details of the UPLC/QE Orbitrap MS analysis of compounds in TGW are described in the Appendix A.

### 4.3. Cell Culture

Rat H9c2 cells were procured from the American Type Culture Collection (Manassas, VA, USA) and were cultured in Dulbecco’s Modified Eagle’s Medium (DMEM, Thermo Fisher Scientific, Inc., Waltham, MA, USA) containing 10% fetal bovine serum (FBS) and antibiotics (1% penicillin–streptomycin) at 37 °C with a humidified atmosphere of 95% air and 5% CO_2_. Cells were sub-cultured at 80% confluency and the medium was changed once in two days. All the experiments were performed within 20–40 passages in order to confirm cell population, uniformity, and reproducibility. The BA was treated 30 min prior to DOX exposure and subsequent stimulation with DOX for 24 h. The cells were then harvested and used in the following experiments.

### 4.4. Animals and Treatment

Male ICR mice were purchased from the Chengdu Da Shuo Experimental Animal Co., Ltd. (Chengdu, China). Mice were housed in standard plastic cages in animal quarters under controlled temperature (22 °C ± 3 °C) in a 12:12 h light-dark cycle, with free access to food and water. TGW pretreatment (100 or 200 mg/kg) was administered for 7 days, then ISO (30 mg/kg body weight) dissolved in saline was injected subcutaneously for 7 consecutive days. The doses administrated were comparable to those used in mouse studies when normalized by body surface area. The control group were fed a normal diet and received subcutaneous injection of PBS (*n* = 6). The ISO group had a normal diet and received subcutaneous injection of ISO (30 mg/kg·day, *n* = 6). The TGW-L group had a normal diet, with TGW treatment at 100 mg/kg·day and subcutaneous injection of ISO (*n* = 6). The TGW-H group had a normal diet, with TGW treatment at 200 mg/kg·day and subcutaneous injection of ISO (*n* = 6). The ratios of heart weight to body weight and left ventricle to body weight were used as measures of cardiac hypertrophy. The ratios of heart weight to body weight ratio and left ventricular weight to body weight were measured as follows. At the end of the experiment, the heart was excised and weighed in its entirety. It was then separated into left ventricle, right ventricle, left atrium, right atrium, and septum, with each component weighed individually. The ratio of heart weight to body weight ratio was calculated by dividing the whole heart weight by the body weight, while the ratio of left ventricular weight to body weight ratio was determined by dividing the left ventricular weight by the body weight. The mice were anesthetized with 4% isofluorane using an Anesthesia Tabletop Bracket with a N_2_O-O_2_ Flow Meter System (Harvard Apparatus, Small Animal Ventilator, Holliston, MA, USA) and were sacrificed by incision of the abdominal artery. Isoflurane has a depressant effect on heart rate and blood pressure, though these effects are typically mild during anesthesia for euthanasia. In rodent models, isoflurane has been shown to induce slight changes in heart function, such as reduced cardiac output, but these are transient and generally reversible once the anesthesia is stopped. However, it is unlikely to cause significant long-term cardiotoxicity in the short time frames associated with euthanasia procedures. Some studies suggest that isoflurane may also be less stressful for animals compared with carbon dioxide, making it a preferable option in certain cases [57,58]. All experimental procedures were carried out in accordance with the National Institute of Health guidelines for the care and use of laboratory animals and were approved by the Institutional Animal Care and Utilization Committee for Medical Science of Wonkwang University (Approved number: WKU 20-115).

### 4.5. Cell Viability and Cell Index Assay

Cell viability was determined using a modified 3- (4,5-dimethylthiazol-2-yl) -2,5-diphenyl tetrazolium (MTT) assay. Briefly, the H9c2 cardiomyocytes were plated in 96-well plates with a cell population of 1 × 10^4^ cells/well and left for attachment overnight at 37 °C in CO_2_ incubator. H9c2 cells were pre-treated with TGW (1, 5 and 10 μg/mL) for 60 min and DOX (1 μM) for 24 h. Next, 10 μL of MTT solution (1 mg/mL) was added to each well and incubated for 4 h in a CO_2_ incubator. After 4 h, the formazan crystals’ precipitate was dissolved by adding 100 μL of dimethyl sulfoxide (DMSO). The absorbance at 540 nm was measured using a spectrofluorometer (F-2500, Hitachi, Tokyo, Japan).

### 4.6. Cell Surface Area Measurement 

The cell surface area, as a hallmark of hypertrophy, was quantified using *F-*actin staining. In brief, H9c2 cells were fixed with 4% paraformaldehyde and 0.1% Triton X-100 was then added to the cells at room temperature for 5 min. Afterwards, cells were blocked with 1% BSA for 1 h. After washing with PBS three times, the cells were stained using phalloidin-Alexa 488 for 30 min to visualize *F*-actin and then counterstained with DAPI. The cells were imaged with a fluorescence microscope and the cell size was analyzed using EVOS-M5000 Cell Imaging System (Thermo Fisher Scientific, Waltham, MA, USA).

### 4.7. Western Blot Analysis 

In order to mitigate potential sampling bias in both the Western blot experiments, random sampling of H9c2 cells was employed. This ensured that the selected samples were representative of the population, thus reducing the risk of skewed data. Additionally, all experiments were repeated at least three times to ensure consistency and reliability. After treatment as indicated above, cells were harvested and washed with cold phosphate-buffered saline (PBS) and followed by incubation in radioimmunoprecipitation (RIPA) buffer at 4 °C for at least 30 min. The cell lysates were clarified by centrifugation at 13,000 rpm for 10 min at 4 °C and the supernatants were collected. The cells’ lysates (35 μg) were separated via 10% SDS-polyacrylamide gel electrophoresis and transferred to nitrocellulose paper. Blots were then washed with H_2_O, blocked with 5% skimmed milk powder in TBST [10 mM Tris-HCl (pH 7.6), 150 mM NaCl, 0.05% Tween-20] for 1 h and incubated with the appropriate primary antibody at dilutions recommended by the supplier. Then, the membrane was washed, primary antibodies were detected with goat anti-rabbit-IgG conjugated to horseradish peroxidase, and the bands were visualized via enhanced chemiluminescence (Amersham, Buckinghamshire, UK). Protein expression levels were determined by analyzing the signals captured on the nitrocellulose membranes using the iBright™ FL1000 (Thermo Fisher scientific, Waltham, MA, USA).

### 4.8. Real-Time PCR

In order to mitigate potential sampling bias in both the real-time PCR experiments, random sampling of H9c2 cells was employed. This ensured that the selected samples were representative of the population, thus reducing the risk of skewed data. Additionally, all experiments were repeated at least three times to ensure consistency and reliability. Cells were collected and RNA extracted using TRIzol^®^ reagent (Invitrogen, Carlsbad, CA, USA). About 1 μg of RNA was reverse transcribed into cDNA using Reverse Transcription Master Premix (ELPISBIO, Republic of Korea). The PCR amplification was conducted with a TOPreal™ qPCR 2X PreMIX (SYBR Green with high ROX, enzynomics) using an Applied Biosystems real-time PCR system (Applied Biosystems, Foster City, CA, USA). Specific sense and antisense primers used were as follows, respectively: ANP, sense: 5′-GAG AAG ATG CCG GTA G-3′, anti-sense: 5′-CTA GAG AGG GAG CTA AGT G-3′; BNP, sense: 5′-TGA TTC TGC TCC TGC TTT TC-3′, anti-sense: 5′-GTG GAT TGT TCT GGA GAC TG-3′; β-MHC, sense: 5′-GCA TTC TCC TGC TGT TTC CT-3′, anti-sense: 5′-CCC AAA TGC AGC CAT CTC-3′; MLC-2v, sense: 5′-CCT AAC GTC ACC GGC AAC C-3, anti-sense: 5′-TTT GGT TCA CAT CAC CCA-3′; GATA-4, sense: 5′-GGG CGA GCC TGT TTG CAA TG-3′, anti-sense: 5′-TGC TTG GAG CTG GCC TGT GA-3′; Bax, sense: 5′-AGA CAC CTG AGC TGA CCT TGG A-3′, anti-sense: 5′-CGC TCA GCC GCT CAG CTT GGT GGA T-3′; Bcl-2, sense: 5′-GGT TGA GCT TGT CCT TCC AG-3′, anti-sense: 5′-CCA TGA CTG AGG GAC CAA CT-3′; caspase-3, sense: 5′-CTC TTC CTG CCC TTC CT-3′, anti-sense: 5′-AGG CCG ACT TCC TGT ATG CT-3′; TGF-beta, sense: 5′-ATC CTG TCC AAA CTA AGG CTC G-3′, anti-sense: 5′-ACC TCT TTA GCA TAG TCC GC-3′; a-SMA, sense: 5′-GAG GCA CCA CTG AAC CCT AA-3′, anti-sense: 5′-CAT CTC CAG AGT CCA GCA CA-3′; collagen I, sense: 5′-ACT TCG TCT CAT TGG GGA TG-3′, anti-sense: 5’-GGA GCA GTA TTC GCC ATC AT-3′; α-tubulin, sense: 5′-GAC CAA GCG TAC CAT CCA GT-3′, anti-sense: 5′-CCA CGT ACC AGT GCA CAA AG-3′. The data are presented as the fold change in gene expression normalized to the endogenous reference gene (α-tubulin) and relative to the untreated control (Table 1).

### 4.9. Immunofluorescence Microscopy

H9c2 cells on glass coverslips were fixed in 4% paraformaldehyde for 30 min and then permeabilized with 0.1% Triton X-100 for 5 min in PBS, followed by washing 3 times with PBS. Non-specific binding of the fixed cells was blocked with PBS containing 1% BSA at 37 °C for 30 min. Next, samples were incubated with primary antibody (GATA-4 and NFAT-3) at 4 °C overnight. Corresponding secondary antibodies were labeled with Alexa Fluor 488 (1:200; Molecular Probes, Eugene, OR, USA). DAPI was used for nuclear staining. The cells were then observed under a fluorescence microscope EVOS-M5000 Cell Imaging System (Thermos fisher scientific, Waltham, MA, USA).

### 4.10. Histopathological Analysis

Left ventricles from each group were randomly selected, and euthanasia was performed by isoflurane. All mice in this study were euthanized using isoflurane to ensure high rigor and reproducibility. The samples were taken from the left ventricles from different groups and fixed with 10% formal saline for 24 h. Washing was carried out with tap water, and serial dilutions of alcohol were then used for dehydration. Specimens were cleared in xylene and embedded in paraffin at 56 °C in a hot water oven for 24 h. Paraffin–beeswax tissue blocks were prepared for sectioning at 4 μm thickness using the slide microtome. The obtained tissue sections were collected on glass slides, deparaffinized, stained by hematoxylin and eosin for routine examination and a Picrosirius Red Stain Kit (Picrosirius Red stain, Polysciences) for histopathological comparisons, determined by light microscopy (EVOSTM M5000, Thermo Fisher Scientific, Bothell, WA, USA). Myocyte cross-sectional areas were measured using the FITC-conjugated WGA (Invitrogen, Thermo Fisher Scientific) -stained sections.

### 4.11. Flow Cytometry of Apoptosis

Apoptosis was measured using an annexin V-FITC/PI double-staining apoptosis detection kit (Cayman Chemicals, Ann Arbor, MI, USA) and flow cytometry. In brief, the H9c2 cells were collected by trypsinization and centrifuged at 1200 rpm for 5 min. Following suspension in binding buffer, cells were labelled with annexin V-FITC and PI for 15 min at 37 °C in the dark. Finally, cell samples were quantified by Attune™ NxT Flow Cytometer (Thermo Fisher Scientific, Waltham, MA, USA). The index of apoptosis was expressed as the percentage of total apoptotic cells, which included the percentage of early apoptotic cells (annexin V-positive and PI-negative) plus the percentage of late apoptotic cells (annexin-V positive and PI-positive).

### 4.12. Statistical Analysis

All the experiments were repeated at least three times. The results were expressed as means ± S.E., and the data were analyzed using one-way ANOVA followed by Dunnett’s test or Student’s *t*-test to determine any significant differences. *p* < 0.05 was considered as statistically significant. All statistical analyses were conducted using SigmaPlot 10.0 (SPSS Inc., Chicago, IL, USA).

## 5. Limitations

This study has several limitations. The H9c2 cell line, while exhibiting certain cardiomyocyte characteristics, does not fully replicate primary cardiomyocytes. Future research should involve the use of primary neonatal cardiomyocytes or in vivo models to address these limitations. In addition, analysis of the components of TGW was carried out, but it was not confirmed which component was responsible for the protective effect against pathological hypertrophy of the heart. This study serves as a preliminary investigation into the effects of TGW, a traditional remedy used historically for heart-related diseases, on cardiac hypertrophy and fibrosis, utilizing both cell and animal models to explore various pathways of cardiac damage. However, a detailed analysis of the underlying mechanisms has not been included, which represents a major limitation of the study. Specifically, the lack of detailed mechanistic analysis of TGW’s actions highlights the need for further research to elucidate these mechanisms. Additionally, the exclusive use of male mice in this study also represents an important limitation. While this choice was made to avoid potential confounding effects of hormonal fluctuations in female mice and to ensure consistency, it may have limited the external validity of our findings. Future studies should include both male and female mice to account for sex-specific differences and improve the generalizability of the results. Therefore, further research is needed to elucidate the molecular and cellular mechanisms for the improvement of cardiac hypertrophy and fibrosis and to enhance the generalizability of the findings, including both male and female mice to account for sex-specific differences.

## Figures and Tables

**Figure 1 ijms-25-10573-f001:**
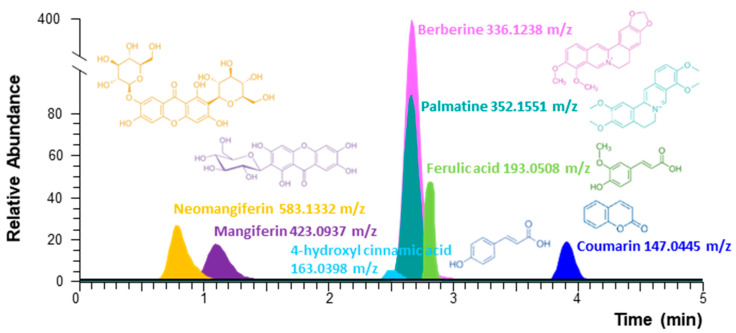
TongGuanWan chemical component analysis. Extracted ion chromatograms (EICs) of UPLC/QE Orbitrap MS analysis of the positive and negative charged molecular ion show seven bioactive compounds.

**Figure 2 ijms-25-10573-f002:**
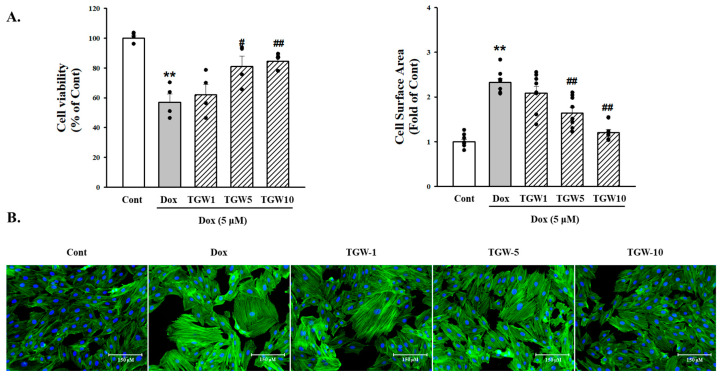
Effects of TongGuanWan on DOX-induced H9c2 cell death and cell surface area size: (**A**) effects of TGW on changes in the viability of H9C2 cells induced by DOX. TGW was pretreated 30 min before DOX administration, and cell viability was assessed using an MTT assay 24 h after DOX treatment; (**B**) representative images of *F-*actin staining (green) and nucleus (blue) in H9c2 cells, used to assess changes in cell surface area size. Representative images and quantitative results demonstrating that TGW (1–10 μg/mL) inhibited DOX (5 μM; 24 h) induced H9c2 cells. All experiments were performed at least three times. Data represent mean ± standard deviation (SD). ** *p *< 0.01, vs. cont; # *p *< 0.05 and ## *p *< 0.01vs. DOX-treated cells.

**Figure 3 ijms-25-10573-f003:**
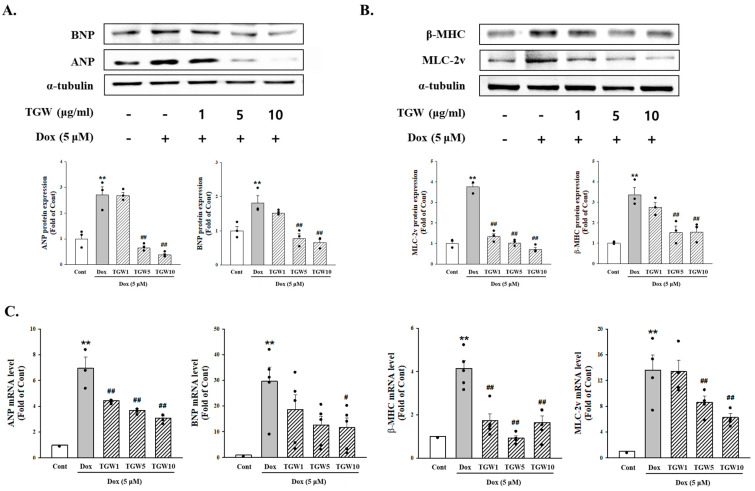
Effects of TongGuanWan on cardiomyocyte hypertrophy markers in DOX-treated H9c2 cells: (**A**) cardiac ANP and BNP protein expression levels and (**B**) β-MHC and MLC-2v protein expression levels and experiments were performed independently three times; (**C**) cardiac signaling pathway responses to TGW determined by RT-qPCR assay. Data are presented as the mean ± SD. ** *p *< 0.01, vs. cont; # *p *< 0.05 and ## *p *< 0.01 vs. DOX-treated cells. TGW, TongGuanWan; ANP, atrial natriuretic peptide; BNP, brain/B-type natriuretic peptide.

**Figure 4 ijms-25-10573-f004:**
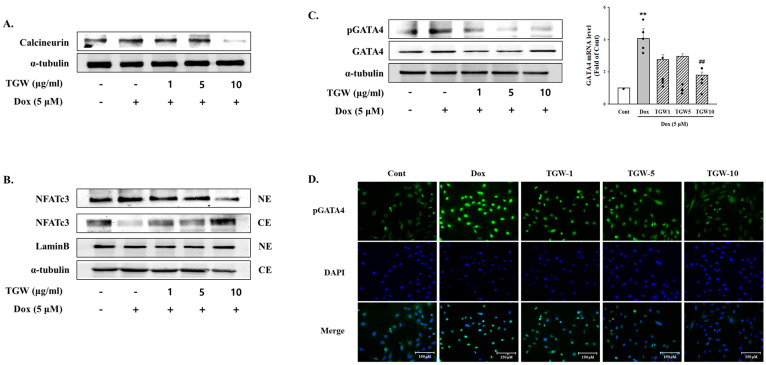
The effects of TongGuanWan on calcineurin-NFAT and -GATA4 signaling pathway signaling protein expression levels in H9c2 cells. The protein levels of calcineurin (**A**), NFATc3 (**B**), GATA4 (**C**), and phosphorylated GATA4 were determined by Western blot analysis. GATA4 mRNA levels were analyzed using real-time PCR. (**D**) Immunofluorescent images of p-GATA-4 nuclear translocation under the laser scanning confocal microscopy are show. Nuclei were stained with DAPI (blue) and p-GATA-4 was stained with Alexa Fluor 488 (green) (immunofluorescence, 200x). The results are expressed as the mean ± SD values of three experiments. ** *p *< 0.01, vs. cont; ## *p *< 0.01 vs. DOX-treated cells.

**Figure 5 ijms-25-10573-f005:**
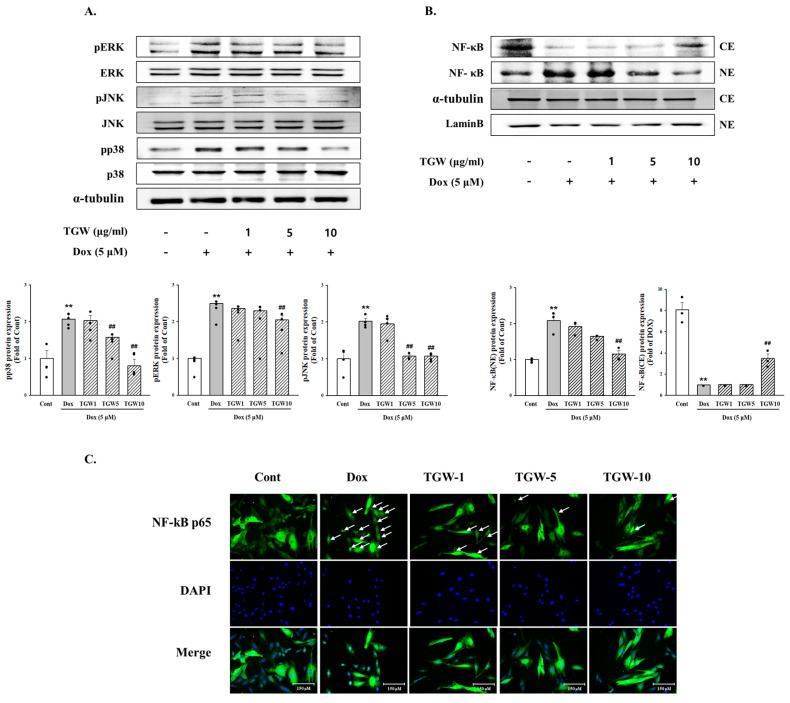
The effects of TongGuanWan on the MAPK/NF-κB signaling pathway in H9c2 cells under DOX conditions: (**A**) protein expression of MAPK (JNK, ERK, and p38) signaling pathway according to Western blot analysis; (**B**) expression and translocation of NF-kB p65 were evaluated by measuring protein in cytoplasmic and nuclear fractions; (**C**) representative images of immunofluorescence staining for NF-κB p65 in H9c2 cells. Immunofluorescence staining used anti-NF-κB p65 (green) and DAPI (blue, nuclei). The white arrow indicates that NF-κB has moved to the nucleus (immunofluorescence, 200×). The results are expressed as the mean ± SD values of three experiments. ** *p *< 0.01, vs. cont; ## *p *< 0.01, vs. DOX-treated cells.

**Figure 6 ijms-25-10573-f006:**
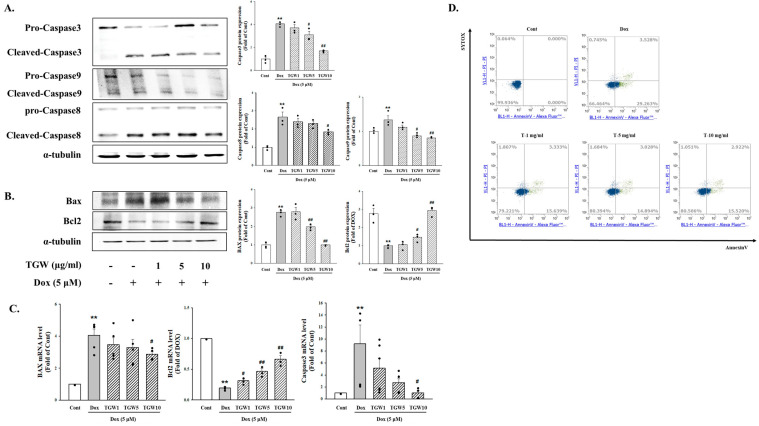
Effects of TongGuanWan on DOX-induced cell apoptosis in H9c2 cells. Western blot analysis results showing the expression levels of apoptosis-related proteins including (**A**) caspase-3, caspase-9, (**B**) Bax and Bcl-2 in DOX-induced H9c2 cells. (**C**) mRNA levels of Bax, Bcl-2, and caspase-3 were quantified using real-time PCR. (**D**) Cellular apoptosis was evaluated using annexin V/PI dual staining followed by flow cytometric analysis. All experiments were conducted in triplicate. Data are presented as mean ± SD. ** *p *< 0.01, vs. cont; # *p *< 0.05 and ## *p *< 0.01, vs. DOX-treated cells.

**Figure 7 ijms-25-10573-f007:**
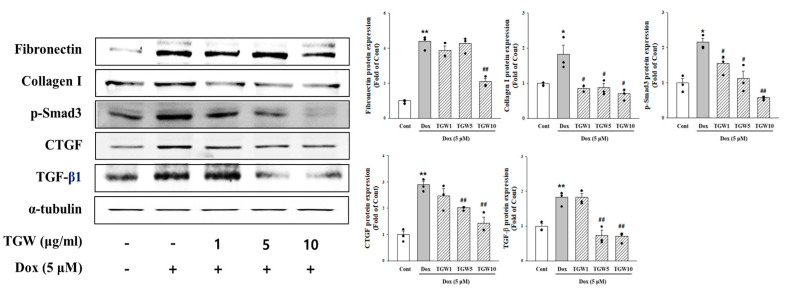
Effects of TGW on cardiac fibrosis markers in H9c2 cells subjected to DOX treatment. Levels of fibronectin, collagen I, α-SMA, TGF-β1, and p-smad3 was determined using Western blot analysis. Data represent mean ± SD. * *p *< 0.05 and ** *p *< 0.01, vs. control; # *p *< 0.05 and ## *p *< 0.01, vs. DOX-treated cells.

**Figure 8 ijms-25-10573-f008:**
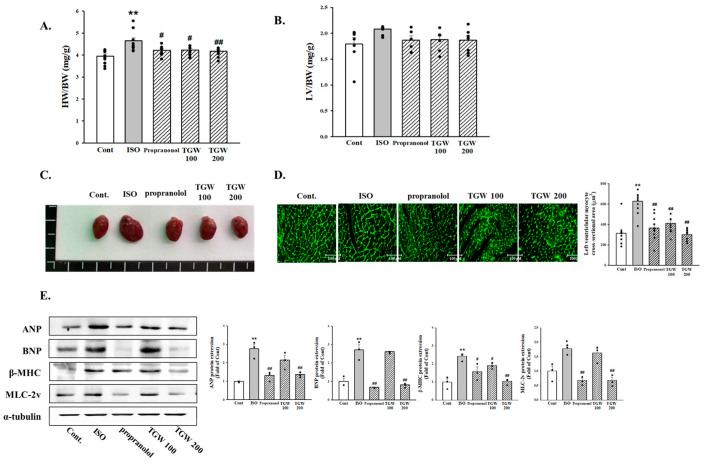
Effects of TongGuanWan on isoproterenol-induced cardiac hypertrophy. TGW administration was started 1 week after infusion of ISO (7 days) in ICR mice. Effects of TGW on heart weight/body weight (**A**) and left ventricular/body weight (**B**) in ISO-induced ICR mice; (**C**) images of the whole hearts of ICR mice; (**D**) WGA staining was performed to evaluate the cell size in heart sections. Green color represents the cardiomyocyte boundary; (**E**) effects of TGW on cardiomyocyte hypertrophy markers in left ventricular tissues. Mice were infused with saline (control), ISO, propranolol (PC), TGW 100 mg/kg·day with ISO (TGW-100), or TGH 200 mg/kg·day with ISO (TGW-200) for 1 week. Data are expressed as mean ± SD. Experimental cases numbered 6~9. * *p *< 0.05 and ** *p *< 0.01 vs. Cont; # *p *< 0.05 and ## *p *< 0.01 vs. ISO group. HW, heart weight; BW, body weight; LV, left ventricular weight; Iso, isoprenaline; Con, control.

**Figure 9 ijms-25-10573-f009:**
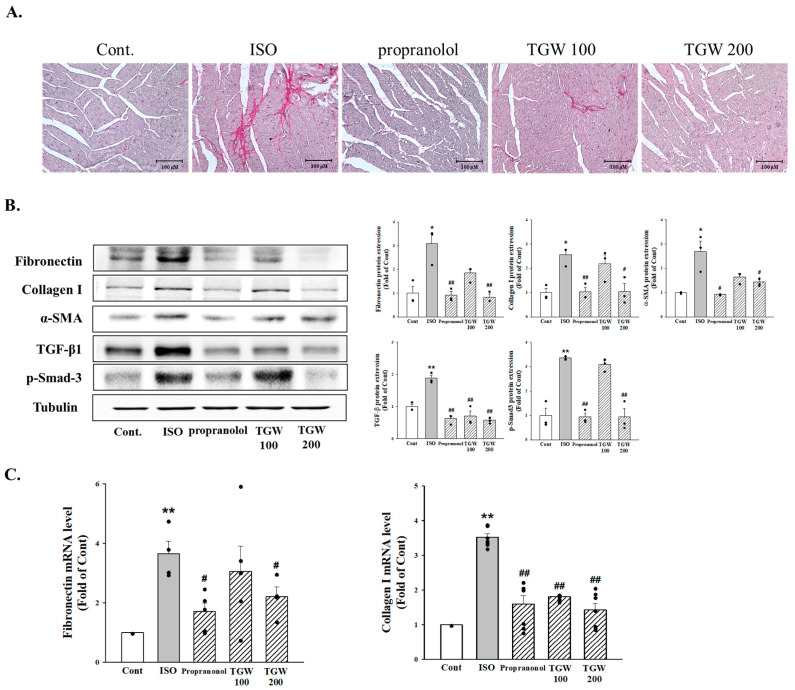
Effects of TongGuanWan on cardiac fibrosis: (**A**) picro-sirius red staining for cardiac fibrosis (red) in a section of the heart; (**B**) representative blots of fibronectin, collagen I, α-SMA, transforming growth factor-β1, and p-Smad3 in the heart tissues of mice in the indicated groups; (**C**) the mRNA expression of markers of fibrosis (Fibronectin and collagen I) was determined by RT-qPCR. Levels of fibronectin, collagen I, α-SMA, TGF-β1, and p-Smad3 was determined using Western blot analysis. Data represent mean ± SD. * *p *< 0.05 and ** *p *< 0.01, vs. control; # *p *< 0.05 and ## *p *< 0.01, vs. ISO group. Iso, isoprenaline; Con, control.

**Table 1 ijms-25-10573-t001:** Primer sequences used in this study.

Target mRNA	Forward Primer	Reverse Primer
ANP	GAGAAGATGCCGGTAG	CTAGAGAGGGAGCTAAGTG
BNP	TGATTCTGCTCCTGCTTTTC	GTGGATTGTTCTGGAGACTG
B-MHC	GCATTCTCCTGCTGTTTCCT	COCCAAATGCAGCCATCTC
ML.C-2v	CCTAACGTCACCGGCAACC	TTTGGTTCACATCACCCA
GATA-4	GGGCGAGCCTGTTTGCAATG	TGCTTGGAGCTGGCCTGTGA
Bax	AGACACCTGAGCTGACCTIGGA	CGCTCAGCCGCTCAGCTTGGTGGAT
Bel-2	GGTTGAGCITGTCCTICCAG	CCATGACTGAGGGACCAACT
Caspase3	СТСТССТСССТТССТ	AGGCCGACTTCCTGTATGCT
TGF-beta	ATCCTGTCCAAACTAAGGCTCG|	ACCICTTTAGCATAGTCCGC
a-SMA	GAGGCACCACTGAACCCTAA	CAICICCAGAGICCAGCACA
Collagen I	ACTICGTCTCATTGGGGATG|	GGAGCAGTATTCGCCATCAT
α-tubulin	GACCAAGCGTACCATCCAGT	CCACGTACCAGTGCACAAAG

## Data Availability

Data are contained within this article.

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
