# Peer review of "TongGuanWan Alleviates Doxorubicin- and Isoproterenol-Induced Cardiac Hypertrophy and Fibrosis by Modulating Apoptotic and Fibrotic Pathways"

_ijms, 2024, doi:10.3390/ijms251910573_

Round 1

Reviewer 1 Report (Previous Reviewer 2)

Comments and Suggestions for Authors

1.      Figure 1 should be checked for accuracy of the data presented

2.      In figure 4C, the white arrow(s) referred to in line 199 must be inserted

3.      The image of figure 7A does not exist. The manuscript does not refer to image 7 B

Author Response

Reviewer 2 Report (New Reviewer)

Comments and Suggestions for Authors

The authors should be very cautious to not self-plagiarize given the eerily similar prior publication:

"TongGuanWan Protects Cardiac Hypertrophy through regulation of Apoptosis and Fibrosis." https://doi.org/10.1096/fasebj.2021.35.S1.02239

Furthermore, this reviewer requests that the authors give a point-by-point explanation of how this present submission differs from the afore mentioned publication, given that the content of the abstracts are functionally identical.

Round 2

Reviewer 2 Report (New Reviewer)

Comments and Suggestions for Authors

This reviewer appreciates the authors' explanation regarding the prior publication, and the expansion of data/findings in the present submission. Nonetheless, the Author Instructions are clear on the matter: "Your manuscript should not contain any information that has already been published." Note that the definition of published means available to public/scientific community view in a journal, server, library, and does not account for the method (example: meeting abstracts).

Please reference: https://www.mdpi.com/journal/ijms/instructions#ethics

Thus, this reviewer requests that the authors revise/reword the structure and content of the abstract and body of the manuscript to avoid any self-plagiarism violations and allow this reviewer to proceed with the evaluation of the scientific rigor of the submission.

Round 3

Reviewer 2 Report (New Reviewer)

Comments and Suggestions for Authors

1.

a) "Abstract: The abstract should be a total of about 200 words maximum. The abstract should be a single paragraph and..."

https://www.mdpi.com/journal/ijms/instructions

b) Furthermore, it would be ideal to state the sample/cohort sizes (n=_) in the abstract, as well as clarify any relevant p-values.

2. In the abstract, the authors claim that based on their findings, "TGW holds potential as a therapeutic option for preventing and treating cardiac hypertrophy and fibrosis..." (line 40) however it is not clear to this reviewer what evidence is presented to support the claim of "prevention", given none of the studies were designed appropriately to defend this claim.

3. The first paragraph could be improved for clarify/accuracy, given "various pathological stimuli" (line 49) does not include endogenous isoproterenol.

4. Why were exclusively male mice utilized? (line 500)

5. A limitations section is necessary for this article.

6. Please clarify how sampling bias was mitigated for both the "4.7 Western Blot Analysis" (line 542) and the "4.8 Real-time PCR" (line 556) to prevent inaccurate representations and subsequently flawed conclusions.

7. Please elaborate on the potential cardiotoxic effects of isoflurane (line 591). Furthermore, clarify if all mice were euthanized by this method, to ensure high rigor and reproducibility with regard to experimental techniques and consistency.

8. For all bar graphs (example: Figure 2A, Figure 3C, et cetera), please provide the scatter plots of each individual point, in addition to the means with respective standard error margins.

9. Please elaborate on why Picro-sirius red as opposed to a trichrome or pentachrome stain was utilized to provide a qualitative assessment of fibrosis/collagen content.

10. Please provide cross-section/short-axis histopathological micrographs of the hearts and/or left ventricles, as depicted in Figure 3 of this manuscript: 

Chinyere IR, Bradley P, Uhlorn J, Eason J, Mohran S, Repetti GG, Daugherty S, Koevary JW, Goldman S, Lancaster JJ. Epicardially Placed Bioengineered Cardiomyocyte Xenograft in Immune-Competent Rat Model of Heart Failure. Stem Cells Int. 2021 Jul 24;2021:9935679. doi: 10.1155/2021/9935679. PMID: 34341667; PMCID: PMC8325579.

11. It would be ideal to provide a graphic/illustration of the study design, to assist in comprehension/contextualization of the data.

Round 4

Reviewer 2 Report (New Reviewer)

Comments and Suggestions for Authors

This reviewer thanks the authors for their thorough revisions to the manuscript. Outside of the missed/mistaken response to critique #11, all changes have been found to be acceptable.

Of note, though the use of a single sex for an animal model cohort/study is pervasive in cardiovascular science for the mentioned reasons (and certainly has historical precedent, increases consistency, et cetera), it still represents a limitation as the findings are limited in external validity.

Author Response

This manuscript is a resubmission of an earlier submission. The following is a list of the peer review reports and author responses from that submission.

Round 1

Reviewer 1 Report

Comments and Suggestions for Authors

This is a study that show TongGuanWan Alleviates DOX and ISO-Induced Cardiac Hypertrophy and Fibrosis by Modulating Apoptotic and Fibrotic 3 Pathways. Concerns for this study:

1.      The author mentioned, “For the evaluation of TGW pretreatment on DOX-induced cytotoxicity, H9c2 cells were pretreated for 60 min with TGW (1-10 μM),” but then mentioned, “However, pretreatment with TGW 10 μg/ml blocked the DOX-induced increase in cell size (3.03±0.17 versus 1.83±0.12 (P<0.01), Figure 2B and C).” Which concentration is correct?

2.      The author mentioned that “H9c2 cells were pretreated for 60 min with TGW (1-10 μM), then the medium was changed, and the cells were treated with DOX (1 μM) for 24 hr.” Is there any difference if, after pretreating for 60 min with TGW, the author did not change the medium before adding DOX?

3.      In Fig 3C, the bar graph for ANP should not be cut unless there are significant fold changes to illustrate.

4.      In Fig 4A, the Western blot band for phospho-ERK in cells treated with DOX does not align with the fold change shown in the bar graph. Which phospho-ERK is being referred to? There are several types of ERK.

5.      Are there any changes in p-NF-κB?

6.      Why is there only one band for NF-κB in the cytoplasmic extract (CE) but two bands in the nuclear extract (NE)?

7.      In Fig 4C, why is the nuclear size different? The nuclear size in TGW-10 is different from DOX, TGW-1, and TGW-5.

8.      In Fig 6A, it should be pro-caspase, not pre-caspase. Why does the author show caspase 8 but not provide data for pro-caspase 8? The data for cleaved caspase 9 is unclear, and the reader is not convinced that such a small change is statistically significant.

9.      Please provide a better and higher magnification of Fig 6D.

10.   In Fig 7D, there is no quantitative data showing significant size differences.

11.   In Fig 8B, the author analyzes SMAD3, but in Fig 8D, the author analyzes p-SMAD3. This is very confusing.

12.   This study does not analyze the detailed mechanisms. Instead, the author uses many different pathways that have been reported. Using so many signaling pathways without showing the exact pathway is not effective.

Comments on the Quality of English Language

Acceptable

Reviewer 2 Report

Comments and Suggestions for Authors

I congratulate the authors for their work. After analyzing this manuscript, I make the following observations and recommendations:

1.      In the caption of figure 1 (lines 159-161), the authors mention 7 bioactive components, but only 6 are presented in the figure. Also, in the Results section (lines 134-135), the authors mention a list of components in a supplementary table, which it is not attached to the manuscript.

2.      in paragraph 2.2 (lines 136-153) the authors refer to figure 2A, 2B and 2C. Picture 2C does not exist in figure 2. Also, in figure 2 there is no clear correlation between the results presented in pictures B-Cont, B-TGW10 and the graph A- Cell surface area (the cells in picture B-TGW10 should have a surface 1.8 times larger than those in picture B-Cont, but both pictures contain an approximately equal number of cells).

3.      In the legend of figure 4C, the authors must mention what the white arrows mean, and check if these arrows correctly mark the same structures in all pictures

4.      In paragraph 2.6 (lines 251-272) the authors talk about the results of pretreatment with TGW, represented graphically in figure 7A, but according to the legend of figure 7A, this image shows the effect of treatment with TGW on ISO-induced ICR mice. Also, the authors present in picture 7B the effect of TGW on left ventricular/Body weight in ISO-induced ICR mice. Can they specify in the Materials and methods section how they measured the left ventricular/Body weight ratio? In line 260 the authors must specify (reference).

5.      The paragraph from lines 428-434 must be revised and reformulated.

Round 2

Reviewer 1 Report

Comments and Suggestions for Authors

The author still has not addressed many of my concerns